# 🌟 MIGA: MIXTURE-OF-EXPERTS WITH GROUP AGGREGATION FOR STOCK MARKET PREDICTION

## ABSTRACT

Stock market prediction has remained an extremely challenging problem for many decades owing to its inherent high volatility and low information noisy ratio. Existing solutions based on machine learning or deep learning demonstrate superior performance by employing a single model trained on the entire stock dataset to generate predictions across all types of stocks. However, due to the significant variations in stock styles and market trends, a single end-to-end model struggles to fully capture the differences in these stylized stock features, leading to relatively inaccurate predictions for all types of stocks. In this paper, we present MIGA, a novel Mixture of Expert with Group Aggregation framework designed to generate specialized predictions for stocks with different styles by dynamically switching between distinct style experts. To promote collaboration among different experts in MIGA, we propose a novel inner group attention architecture, enabling experts within the same group to share information and thereby enhancing the overall performance of all experts. As a result, MIGA significantly outperforms other end-to-end models on three Chinese Stock Index benchmarks including CSI300, CSI500, and CSI1000. Notably, MIGA-Conv reaches 24 % excess annual return on CSI300 benchmark, surpassing the previous state-of-the-art model by 8% absolute. Furthermore, we conduct a comprehensive analysis of mixture of experts for stock market prediction, providing valuable insights for future research.

## 1    INTRODUCTION

The stock market, a complex dynamic system, has long attracted investors worldwide who seek to generate high returns through risk-taking and achieve their investment objectives. However, due to the inherent volatility and noisy characteristics of financial markets (Fama, 1970; Malkiel, 2003), forecasting stock prices accurately and making profitable investment decisions are extremely challenging tasks. These challenges are considered to stem from the interplay of multiple stock factors, including market participants' behavior (Shiller, 2003), macroeconomic variables (Chen, 1988), and the difficult-to-quantify flow of information (Grossman & Stiglitz, 1980).

Over past decades, machine learning (ML) and deep learning (DL) methods have shown remarkable capabilities in stock market prediction by learning from a dataset comprised of various stock factors in an end-to-end manner. Most existing ML or DL methods adopt an advanced network architectures derives from some basic networks like LSTM (Hochreiter, 1997) and Transformer (Vaswani et al., 2017). For instance, FactorVAE (Duan et al., 2022) introduces a novel approach to predicting excess returns and benchmark returns separately using Variational Auto-Encoders (Kingma, 2013). MAS-TER (Li et al., 2024) proposes a transformer-based model that addresses complex stock correlations through alternating intra-stock and inter-stock information aggregation. These methods adopt stock market prediction as a supervised learning task, where the historical stock factors is always seemed as features and the return rate is applied as the target for a single end-to-end model. Therefore, the inherent temporal characteristics of stock market enable end-to-end models to effectively predict future stock returns.

However, in the context of financial markets, stocks of different styles often tend to exhibit significant variations in their features of stock factors. For instance, large-cap stocks are stable and offer predictable growth, appealing to risk-averse investors seeking modest returns. But mall-cap stocks

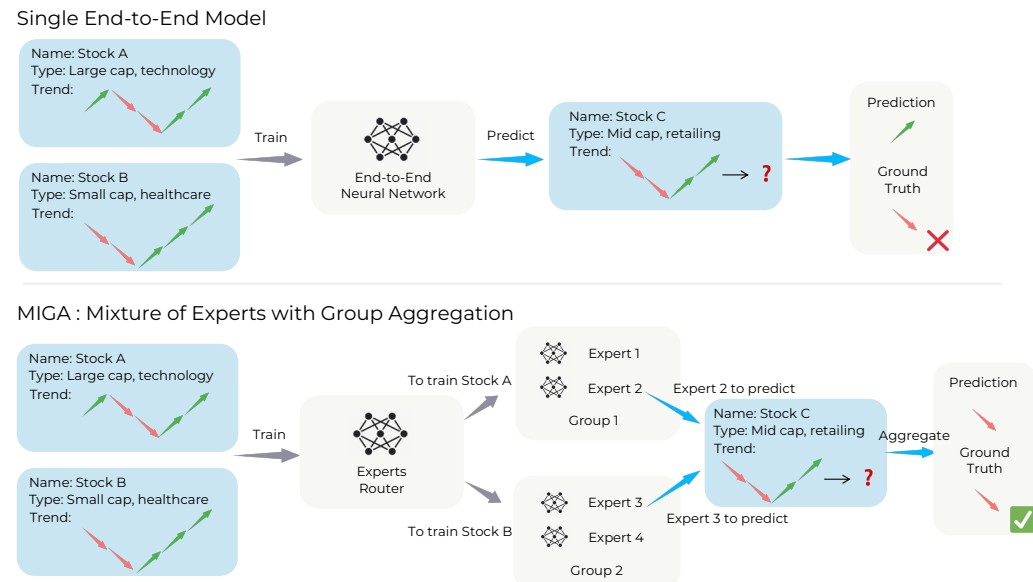

Figure 1: Comparison between MIGA and single end-to-end model. MIGA generates specialized predictions for stocks with different styles by dynamically switching between distinct style experts.

are more volatile and have higher growth potential, making them suitable for investors with a higher risk tolerance and a longer investment horizon. These salient stylized features are often input into the training of a single end-to-end model without differentiation, enabling single end-to-end model struggles to capture the differences in these stylized stock features. In this paper, we introduce MIGA, a novel **Mi**xture of Expert with **G**roup **A**ggregation framework that could generate specialized predictions of various styles of stocks by dynamically switching between specialized experts. As depicted in Figure 1, the proposed MIGA architecture diverges from traditional end-to-end models in its two-stage design. The first stage employs an expert router, built upon the end-to-end model, which encodes raw stock data into vector representations and subsequently converts these representations into expert allocation weights via a linear transformation. The second stage introduces a novel expert gathering aggregation structure, wherein experts are grouped and an internal group attention mechanism is incorporated within each expert group to facilitate knowledge sharing and collaboration among experts, thereby enhancing the overall predictive performance of the model.

To validate MIGA, we conduct a comprehensive evaluation using 3 distinct types of feature encoders (convolution-based, recurrence-based, and attention-based) as our expert router, and linear layer as our experts, consistent with the design principles established in prior research (Dai et al., 2024; Zoph et al., 2022).We denote these three MIGA models as MIGA-Conv, MIGA-Rec, and MIGA-Attn. Our experimental evaluation involves assessing the performance of these MIGA models on three prominent Chinese stock indices (CSI300, CSI500, and CSI1000) with long-only and long-short stock portfolios. The results demonstrate that our MIGA models exhibit superior performance, outperforming all other end-to-end models. Notably, on the CSI300 benchmark, MIGA-Conv achieves a remarkable 24% excess annual return with a long-only portfolio. Furthermore, we provide an in-depth analysis of the Mixture of Expert for stock prediction, providing valuable insights for future work.

## 2 MIGA: MoE WITH GROUP AGGREGATION

A Mixture of Experts (MoE) architecture comprises two main components: the router and the experts. The router assigns weights for each data point, while each expert generates its own prediction. The final output of the MoE is the weighted aggregation of the predictions from all experts. In this section, we present the methodology by which MIGA generates predictions and undergoes training process.

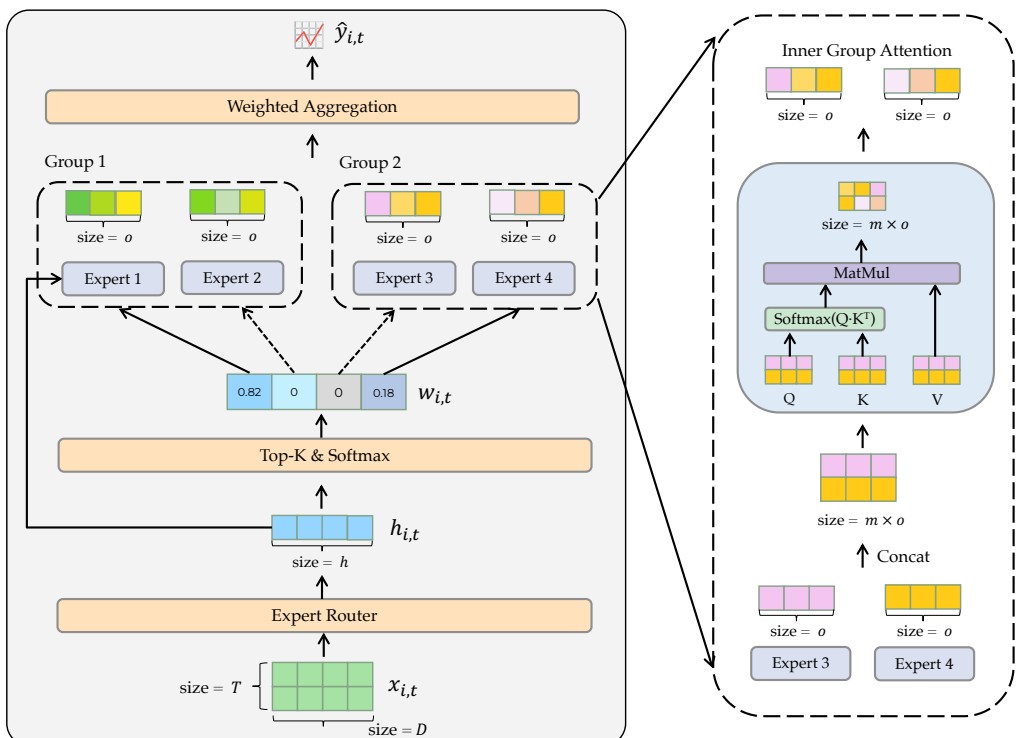

Figure 2: Overview of Mixture of Expert with Group Aggregation. nner group attention refers to the self-attention mechanism within each expert group, which facilitates the aggregation of information among experts within the same group.

## 2.1 PROBLEM FORMULATION

Following previous work (Duan et al., 2022; Li et al., 2024), we formulate stock market prediction as a supervised learning task by generating trading strategies based on daily cross-sectional stock prices. Consider a stock pool of size $N$, where for each stock $i$ on trading day $t$, there is a corresponding average price $p_t^i$ and a feature vector $x_t^i \in \mathcal{R}^{N \times T \times D}$, where $D$ is the number of features and $T$ is the range of days for stocks. For each feature vector $x_t^i$, we utilize the stock's future return rate as a training label to supervise model learning which can be calculated as follows:

$$y_t^i = \frac{p_{t+2}^i - p_{t+1}^i}{p_{t+1}^i} \tag{1}$$

Unless otherwise stated, we use $x_{i,t}$, $y_{i,t}$ to denote the sequential feature vectors $(x_{t-T}^i, ..., x_t^i)$ and future return $y_t^i$ of stock $i$ on time $t$ in the following of this work for simplicity.

## 2.2 ROUTING CROSS GROUP EXPERTS

### 2.2.1 ROUTER

As depicted in Figure 2, we initiate the processing pipeline by leveraging a trainable router to perform cross-sectional encoding of the stock set $\mathcal{S}_{|N|}^t = (x_{1,t}, x_{2,t}, ..., x_{|N|,t})$, thereby capturing the complex inter-relationships between the constituent stocks and generate routing weights for them. Subsequently, we employ a top-k selection strategy to identify the most salient k experts, characterized by the highest routing weights, and then apply a softmax normalization to the selected subset, thereby obtaining the final, normalized weights that capture the relative importance of each expert's contribution. This process can be expressed as:

$$\mathcal{H}_{\mathcal{G} \times \mathcal{E}}^{i,t} = \mathbf{Router}(x_{i,t}) \tag{2}$$

$$\mathcal{H}_{\mathcal{G} \times \mathcal{E}}^{i,t} = \{h_{j,k}^{i,t}\}, j = 1, 2, ..., |\mathcal{G}|, k = 1, 2, ..., |\mathcal{E}| \tag{3}$$

$$\{w_{j,k}^{i,t}\}_{\mathcal{G} \times \mathcal{E}} = \begin{cases} \text{softmax}(\{h_{j,k}^{i,t}\}), & h_{j,k}^{i,t} \in \textbf{TopK}(\mathcal{H}_{\mathcal{G} \times \mathcal{E}}^{i,t}) \\ 0, & \text{otherwise} \end{cases} \tag{4}$$

where $\mathcal{G}, \mathcal{E}$ indicates the group pool and the expert pool respectively, $h_{j,k}^{i,t}$ denotes the latent state representation of the router's output for the $i$-th stock of day $t$ at the $k$-th of the $j$-th group and $w_{j,k}^{i,t}$ represents the routing weight for the $i$-th stock of day $t$ assigned by the router to each expert at the $k$-th of the $j$-th group, which governs the degree of contribution from each expert to the final output.

### 2.2.2 EXPERT

For expert models, each expert takes the same hidden representation of stocks generated by router as input and generates their own predictions $o_{j,k}^{i,t}$. In MIGA, we set all experts as linear layers to prevent the final model from becoming overly complex. The generation process can be expressed as:

$$o_{j,k}^{i,t} = \textbf{Expert}_{j,k}(h_{j,k}^{i,t}) \tag{5}$$

As illustrated in Figure 1, these predictions are then aggregated through an inner-group attention mechanism, which facilitates interaction among the experts within the group and enriches the output representation of each individual expert through information sharing.

### 2.2.3 GROUP AGGREGATION

For group aggregation, we begin by concatenating the output $o_{j,k}^{i,t}$ representations of all constituent experts to form a unified input representation. Subsequently, we generate the query $Q_j^{i,t}$, key $K_j^{i,t}$, and value $V_j^{i,t}$ matrices necessary for the inner-group attention mechanism as follows:

$$O_j^{i,t} = \textbf{Concat}(\{o_{j,k}^{i,t} | k = 1, 2, ..., |\mathcal{E}|\}) \tag{6}$$

$$Q_j^{i,t} = W_q O_j^{i,t}, K_j^{i,t} = W_k O_j^{i,t}, V_j^{i,t} = W_v O_j^{i,t} \tag{7}$$

The outputs of individual experts within the group are then cross-interacted via the attention mechanism, yielding a mixed output $\bar{O}_j^{i,t} = \{\bar{o}_{j,k}^{i,t} | k = 1, 2, ..., |\mathcal{E}|\}$ representation that aggregates the knowledge and insights of all experts in the group.

$$\bar{O}_j^{i,t} = \textbf{MultiHeadSelfAttention}(Q_j^{i,t}, K_j^{i,t}, V_j^{i,t}) \tag{8}$$

After obtaining the weight $w_{j,k}^{i,t}$ emitted by the router and the aggregated predictions $\bar{o}_{j,k}^{i,t}$ from each expert within the expert group for future returns, we compute the weighted composite prediction of all experts to derive MIGA's prediction of future profitability for stock $i$ at time $t$. And for each prediction $\hat{y}_t^i$, we have:

$$\hat{y}_t^i = \sum_{j=0}^{\mathcal{G}} \sum_{k=0}^{\mathcal{E}} w_{j,k}^{i,t} \bar{o}_{j,k}^{i,t} \tag{9}$$

### 2.3 TRAINING

**Expert Loss** For the training process, we depart from the conventional approach of employing mean squared error (MSE) (Li et al., 2024) as the loss function for model training. Instead, we incorporate the information coefficient (IC) into the loss calculation framework to ensure the correlation between labels and predictions. Consider the actual stock cross-sectional returns $Y_{label}^t = \{y_t^i | i = 1, 2, ..., N\}$ at time $t$ and the predicted stock cross-sectional returns $Y_{pred}^t = \{\hat{y}_t^i | i = 1, 2, ..., N\}$, we calculated the $\mathcal{L}_{\text{Expert}}$ as follows:

$$\mathcal{L}_{\text{Expert}} = -\frac{1}{|\mathcal{T}|} \sum_{t \in \mathcal{T}} \frac{cov(Y_{pred}^t, Y_{label}^t)}{\sqrt{var(Y_{pred}^t)var(Y_{label}^t)}} \tag{10}$$

**Load Balance Consideration** Automatically learned routing strategies are susceptible to load imbalance, which can lead to the phenomenon of routing collapse (Shazeer et al., 2017a). Specifically, this occurs when the model consistently favors a subset of experts, thereby depriving other experts of adequate training opportunities and hindering their ability to contribute to the overall performance of the model. In many existing MoE architectures designed for training large language models (Zoph et al., 2022; Dai et al., 2024), token load balance auxiliary loss (Shazeer et al., 2017a) is employed to ensure that all tokens are distributed as evenly as possible among experts. However, this approach is not directly applicable to stock market prediction tasks, as there is no analogous concept of tokens in stock market. Hence, we reduce this imbalance by minimizing the distance between the router output $h_t^i = \{h_{j,k}^{i,t}\}$ and its mean value. The loss function can be written as:

$$\mathcal{L}_{\text{Router}} = \sum_{t \in \mathcal{T}} \sum_{i=1}^{N} (h_t^i - \text{mean}(h_t^i))^2 \tag{11}$$

Integrating all the aforementioned considerations, we arrive at the final loss function utilized for training MIGA:

$$\mathcal{L}_{\text{MIGA}} = \alpha \mathcal{L}_{\text{Router}} + \beta \mathcal{L}_{\text{Expert}} \tag{12}$$

By minimizing $\mathcal{L}_{\text{MIGA}}$, we can optimize the correlation ($\mathcal{L}_{\text{Expert}}$) between predicted returns and actual returns, concurrently mitigating load imbalance by reducing the routing loss ($\mathcal{L}_{\text{Router}}$).

## 3 EXPERIMENTS

### 3.1 IMPLEMENTATION DETAILS

To instantiate MIGA, we leverage PyTorch to implement the entire MoE architecture and utilize NVIDIA A100-80GB GPU for training. For each model in our experiments, we set the maximum number of training epochs to 60 and implement an early stopping strategy to prevent overfitting and ensure model generalizability. We set the initial learning rate at 5e-4 and the hyper-parameters $\alpha, \beta$ in $\mathcal{L}_{\text{MIGA}}$ at 2e-3, 1 respectively. Following previous work (Li et al., 2024), we set the history stock sequence length $T$ as 5, which is a week of trading days. Furthermore, for the selection of the number of groups and experts, we conduct a comprehensive analysis in Section 3.5.

### 3.2 EVALUATION SETUP

**Datasets** Our dataset comprises 626 daily features which covers all stocks in the Chinese stock market, ranging from January 1, 2014, to July 25, 2024. We partitioned the dataset temporally, using data from January 1, 2014, to July 25, 2022, as the training set, data from July 25, 2022, to July 25, 2023, as the validation set, and data from July 25, 2023, to July 25, 2024, as the test set. At the end of each training epoch, the model was evaluated on the validation set, and the optimal model configuration was retained as the final model.

**Baselines** To evaluate the performance of MIGA more comprehensively, we replaced the stock representation encoder in different routers and trained three different MIGA models. Specifically, for MIGA-Conv, we adopted the convolutional operation from the Temporal Convolutional Network (TCN) (Hewage et al., 2020). For MIGA-Rec, we utilized an LSTM (Greff et al., 2016) as the encoder in our router. For MIGA-Attn, we employed the Transformer (Ding et al., 2020) encoder. Subsequently, We compare the performance of MIGA models (MIGA-Conv, MIGA-Rec, MIGA-Attn) with several stock price forecasting baselines : TCN (Hewage et al., 2020), LSTM (Greff et al., 2016), Transformer (Vaswani et al., 2017) and three previous state-of-the-art (SoTA) models: ModernTCN (Donghao & Xue, 2024), Mamba (Gu & Dao, 2023) and MASTER (Li et al., 2024).

**Benchamrks** We evaluate all MIGA models on three Chinese Stock Index benchmarks including CSI300, CSI500 and CSI1000 stock sets. The CSI300 index is a blue-chip index comprising 300

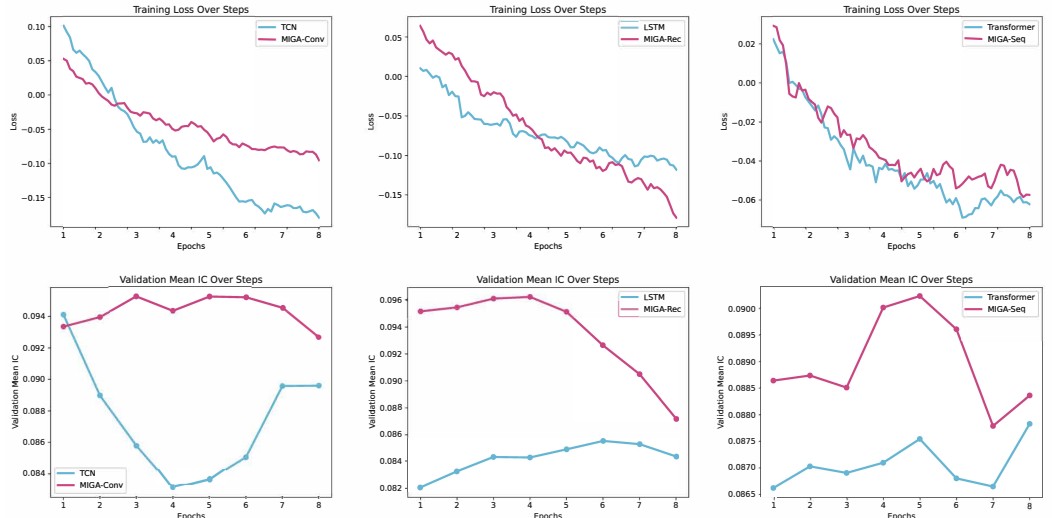

Figure 3: The comparison of the training loss and validation set IC between MIGA and single end-to-end models during the first 8 epochs, where the optimal performance of most models emerges.

large-cap stocks with superior liquidity. In contrast, the CSI500 index is a mid-cap index that includes 500 medium-sized companies with strong liquidity, characterized by their growth potential and dynamic business models, making the CSI500 a valuable indicator of the mid-cap segment of China's A-share market. The CSI1000 index, on the other hand, consists of 1,000 small and medium-cap stocks. Collectively, these three indices provide a comprehensive representation of the overall performance of China's A-share market.

### 3.3 METRICS

To provide a comprehensive assessment of the model's performance, we employ a combination of ranking metrics and portfolio-based metrics. Specifically, we consider four ranking metrics: Information Coefficient (IC), Rank Information Coefficient (RankIC), Information Ratio-based IC (ICIR), and Information Ratio-based RankIC (RankICIR). These metrics are defined as follows: IC and RankIC represent the Pearson correlation coefficient and Spearman correlation coefficient, averaged at a daily frequency. ICIR and RankICIR are normalized variants of IC and RankIC, obtained by dividing by the standard deviation.

$$ICIR = \frac{Mean(IC)}{Std(IC)}, RankICIR = \frac{Mean(RankIC)}{Std(RankIC)} \tag{13}$$

These metrics are widely used in the literature (e.g., Xu et al. (2021); Yang et al. (2020)) to evaluate the performance of forecasting models from both value and rank perspectives.

For portfolio-based metrics, we sort all stocks daily based on their predicted return rate, selecting the top 5% for a long-only strategy. Simultaneously, we identify the bottom 5% for short selling, combining both positions into a long-short portfolio. Following previous work (Li et al., 2024), we report the excess annualized return (AR) and information ratio (IR) metrics. The AR metric measures the annual expected excess return generated by the investment, while the IR metric evaluates the risk-adjusted performance of the investment.

### 3.4 RESULT

Table 1, 2 present the results of MIGA on 3 Chinese Stock Index (CSI300, CSI500, CSI1000) benchmarks, highlighting the following salient observations:

**MIGA achieves SoTA performance by surpassing the previous SoTA models.** MIGA models achieve the best results on 15/16 of the ranking metrics and 14/16 portfolio-based metrics. Notably,

Table 1: Overall performance comparison on ranking metrics. For each model, we conducted three experiments, using the mean as the final result and the standard deviation (in brackets) to indicate the confidence interval. The best results are in bold and the second-best results are underlined.

| Stockset | Type | Model | IC | ICIR | RankIC | Rank ICIR |
|---|---|---|---|---|---|---|
| CSI300 | Baselines | TCN | 0.041 (0.001) | 0.272 (0.034) | 0.063 (0.003) | 0.382 (0.043) |
| | | LSTM | 0.013 (0.002) | 0.127 (0.015) | 0.026 (0.005) | 0.249 (0.031) |
| | | Transformer | 0.039 (0.001) | 0.222 (0.017) | 0.066 (0.015) | 0.375 (0.016) |
| | SoTA models | Mamba | 0.041 (0.008) | 0.239 (0.018) | 0.060 (0.016) | 0.333 (0.029) |
| | | MASTER | 0.040 (0.008) | 0.248 (0.003) | 0.065 (0.013) | 0.363 (0.004) |
| | | ModernTCN | 0.041 (0.005) | 0.240 (0.013) | 0.067 (0.011) | 0.363 (0.007) |
| | MIGA | MIGA-Conv | **0.052 (0.002)** | 0.265 (0.007) | **0.079 (0.004)** | 0.365 (0.014) |
| | | MIGA-Rec | 0.034 (0.007) | 0.230 (0.011) | 0.046 (0.011) | 0.297 (0.012) |
| | | MIGA-Attn | 0.048 (0.002) | **0.273 (0.004)** | 0.074 (0.004) | **0.394 (0.010)** |
| CSI500 | Baselines | TCN | 0.035 (0.004) | 0.253 (0.018) | 0.058 (0.004) | 0.388 (0.041) |
| | | LSTM | 0.024 (0.009) | 0.233 (0.061) | 0.039 (0.014) | 0.370 (0.085) |
| | | Transformer | 0.031 (0.003) | 0.236 (0.015) | 0.058 (0.006) | 0.433 (0.038) |
| | SoTA models | Mamba | 0.037 (0.003) | 0.281(0.030) | 0.056 (0.009) | 0.392 (0.036) |
| | | MASTER | 0.034 (0.003) | 0.269 (0.018) | 0.057 (0.006) | 0.413 (0.020) |
| | | ModernTCN | 0.040 (0.004) | 0.299 (0.022) | 0.062 (0.010) | 0.424 (0.011) |
| | MIGA | MIGA-Conv | 0.042 (0.003) | 0.279 (0.027) | **0.069 (0.001)** | 0.404 (0.012) |
| | | MIGA-Rec | **0.042 (0.002)** | **0.326 (0.018)** | 0.049 (0.010) | 0.363 (0.067) |
| | | MIGA-Attn | 0.035 (0.001) | 0.270 (0.017) | 0.059 (0.002) | **0.433 (0.012)** |
| CSI1000 | Baselines | TCN | 0.040 (0.004) | 0.310 (0.025) | 0.067 (0.001) | 0.471 (0.011) |
| | | LSTM | 0.032 (0.006) | 0.302 (0.031) | 0.057 (0.004) | **0.502 (0.010)** |
| | | Transformer | 0.034 (0.002) | 0.250 (0.031) | 0.069 (0.003) | 0.495 (0.005) |
| | SoTA models | Mamba | 0.042 (0.001) | 0.311 (0.018) | 0.059 (0.008) | 0.397 (0.137) |
| | | MASTER | 0.040 (0.001) | 0.311 (0.007) | 0.068 (0.003) | 0.483 (0.165) |
| | | ModernTCN | 0.047 (0.001) | 0.358 (0.015) | 0.071 (0.004) | 0.477 (0.008) |
| | MIGA | MIGA-Conv | 0.043 (0.002) | 0.299 (0.016) | **0.072 (0.003)** | 0.452 (0.004) |
| | | MIGA-Rec | **0.048 (0.001)** | **0.383 (0.017)** | 0.069 (0.001) | 0.496 (0.044) |
| | | MIGA-Attn | 0.039 (0.001) | 0.296 (0.013) | 0.067 (0.001) | 0.493 (0.007) |
| Average | Baselines | TCN | 0.039 (0.009) | 0.278 (0.077) | 0.063 (0.008) | 0.414 (0.095) |
| | | LSTM | 0.023 (0.017) | 0.221 (0.107) | 0.041 (0.023) | 0.374 (0.126) |
| | | Transformer | 0.035 (0.006) | 0.236 (0.063) | 0.064 (0.024) | 0.434 (0.059) |
| | SoTA models | Mamba | 0.040 (0.012) | 0.277 (0.066) | 0.058 (0.033) | 0.374 (0.202) |
| | | MASTER | 0.038 (0.012) | 0.276 (0.028) | 0.063 (0.022) | 0.420 (0.189) |
| | | ModernTCN | 0.043 (0.010) | 0.299 (0.050) | 0.067 (0.025) | 0.421 (0.026) |
| | MIGA | MIGA-Conv | **0.046 (0.007)** | 0.281 (0.050) | **0.073 (0.008)** | 0.407 (0.030) |
| | | MIGA-Rec | 0.041 (0.010) | **0.313 (0.046)** | 0.055 (0.022) | 0.385 (0.123) |
| | | MIGA-Attn | 0.041 (0.004) | 0.280 (0.034) | 0.067 (0.007) | **0.440** (0.029) |

MIGA-Conv achieves 24% improvement in long-only portfolio metrics compared to ModernTCN, the SoTA end-to-end model, with an annual return (AR) of 0.24 versus 0.18 and a Information Ratio (IR) of 1.80 versus 1.37 on CSI300 benchmark as well as a IC of 0.046 versus 0.043 and a Rank IC of 0.073 versus 0.067 on of the average on 3 benchmarks. Although the ICIR and RankICIR are slightly lower, the final portfolio return of MIGA is higher than ModernTCN. The achievements in both types of metrics imply that MIGA is of good predicting ability on the 3 stock set without a trade-off.

**MIGA can significantly enhance the performance of a single end-to-end model.** Compared with the baseline end-to-end models, MIGA significantly outperforms them on all benchmarks. For example, MIGA-Conv and TCN are both convolution-based models, with MIGA-Conv demonstrating superior performance across all CSI benchmarks. For instance, MIGA-Conv substantially outperforms TCN under long-only portfolio (0.24 vs 0.13 on AR, 1.80 vs 0.97 on IR). This trend is also observed in the comparison between MIGA-Rec and LSTM, as well as MIGA-Attn and Transformer, highlighting the architectural advantages of MIGA. Furthermore, compared to single end-to-end models, MIGA

Table 2: Comparison of portfolio result on three stocksets. For each model, we conducted three experiments, using the mean as the final result and the standard deviation (in brackets) to indicate the confidence interval. The best results are in bold and the second-best results are underlined.

| Stockset | Type | Model | Long-only | | Long-short | |
|---|---|---|---|---|---|---|
| | | | AR | IR | AR | IR |
| CSI300 | Baselines | TCN | 0.13 (0.07) | 0.97 (0.50) | 0.60 (0.13) | 3.33 (0.73) |
| | | LSTM | -0.03 (0.05) | -0.18 (0.38) | 0.11 (0.07) | 0.86 (0.58) |
| | | Transformer | 0.17 (0.08) | 1.24 (0.51) | 0.66 (0.22) | 3.64 (0.65) |
| | SoTA models | Mamba | 0.15 (0.10) | 1.06 (0.73) | 0.66 (0.09) | 3.71 (0.91) |
| | | MASTER | 0.16 (0.04) | 1.22 (0.33) | 0.57 (0.09) | 3.49 (0.40) |
| | | ModernTCN | 0.18 (0.08) | 1.37 (0.52) | 0.75 (0.08) | 4.10 (0.33) |
| | MIGA | MIGA-Conv | **0.24 (0.04)** | **1.80 (0.29)** | **0.82 (0.10)** | 3.94 (0.50) |
| | | MIGA-Rec | 0.07 (0.02) | 0.50 (0.02) | 0.55 (0.15) | 3.39 (0.71) |
| | | MIGA-Attn | 0.20 (0.01) | 1.47 (0.03) | 0.80 (0.06) | **4.37 (0.29)** |
| CSI500 | Baselines | TCN | 0.13 (0.05) | 0.73 (0.26) | 1.00 (0.07) | 6.80 (0.34) |
| | | LSTM | 0.08 (0.06) | 0.45 (0.33) | 0.67 (0.26) | 5.21 (1.39) |
| | | Transformer | 0.14 (0.04) | 0.78 (0.19) | 0.91 (0.08) | 5.95 (0.77) |
| | SoTA models | Mamba | 0.17 (0.05) | 0.94 (0.29) | 1.09 (0.26) | 6.92 (2.16) |
| | | MASTER | 0.18 (0.04) | 0.99 (0.23) | 1.03 (0.16) | 6.67 (1.01) |
| | | ModernTCN | 0.14 (0.04) | 0.76 (0.26) | 0.99 (0.13) | 6.51 (0.88) |
| | MIGA | MIGA-Conv | **0.18 (0.03)** | **1.07 (0.18)** | **1.17 (0.03)** | 7.06 (0.03) |
| | | MIGA-Rec | 0.11 (0.06) | 0.64 (0.36) | 1.17 (0.11) | **7.79 (0.51)** |
| | | MIGA-Attn | 0.11 (0.02) | 0.60 (0.13) | 0.89 (0.20) | 6.51 (0.88) |
| CSI1000 | Baselines | TCN | 0.11 (0.02) | 0.51 (0.07) | 1.49 (0.10) | 8.39 (0.47) |
| | | LSTM | 0.13 (0.04) | 0.59 (0.17) | 1.18 (0.12) | 8.40 (0.32) |
| | | Transformer | **0.17 (0.02)** | **0.77 (0.10)** | 1.45 (0.11) | 8.43 (0.82) |
| | SoTA models | Mamba | 0.10 (0.06) | 0.45 (0.26) | 1.55 (0.25) | 9.08 (1.35) |
| | | MASTER | 0.13 (0.03) | 0.60 (0.13) | 1.47 (0.17) | 8.40 (0.72) |
| | | ModernTCN | 0.14 (0.04) | 0.69 (0.21) | 1.67 (0.17) | 9.89 (0.72) |
| | MIGA | MIGA-Conv | 0.15 (0.02) | 0.69 (0.09) | 1.61 (0.12) | 8.76 (0.67) |
| | | MIGA-Rec | 0.13 (0.04) | 0.61 (0.20) | **1.74 (0.10)** | **10.20 (0.48)** |
| | | MIGA-Attn | 0.09 (0.02) | 0.42 (0.08) | 1.41 (0.07) | 8.16 (0.44) |
| Average | Baselines | TCN | 0.12 (0.14) | 0.74 (0.83) | 1.03 (0.30) | 6.17 (1.54) |
| | | LSTM | 0.06 (0.15) | 0.29 (0.88) | 0.65 (0.45) | 4.82 (2.29) |
| | | Transformer | 0.16 (0.14) | 0.93 (0.80) | 1.01 (0.41) | 6.01 (2.24) |
| | SoTA models | Mamba | 0.14 (0.21) | 0.82 (1.28) | 1.10 (0.60) | 6.57 (4.42) |
| | | MASTER | 0.16 (0.11) | 0.94 (0.69) | 1.02 (0.42) | 6.19 (2.13) |
| | | ModernTCN | 0.15 (0.16) | 0.94 (0.99) | 1.14 (0.38) | 6.83 (1.93) |
| | MIGA | MIGA-Conv | **0.19 (0.09)** | **1.19 (0.56)** | **1.20 (0.25)** | 6.59 (1.20) |
| | | MIGA-Rec | 0.10 (0.12) | 0.58 (0.58) | 1.15 (0.36) | **7.13 (1.70)** |
| | | MIGA-Attn | 0.13 (0.05) | 0.83 (0.24) | 1.03 (0.33) | 6.35 (1.61) |

demonstrates a more balanced performance across different benchmarks by aggregating different experts. This approach allows MIGA to effectively adapt to the varying characteristics of the CSI300, CSI500, and CSI1000 indices by switching the expert with different styles, showcasing its effectiveness and versatility in diverse market styles.

**MIGA exhibits superior stock market prediction capabilities on unseen dataset.** As illustrated in Figure 3, we compare the training loss and validation set information correlation (IC) between the Multi-Input Gradient-Aware (MIGA) model and single end-to-end models during the first 8 epochs, a period during which the optimal performance of most models typically emerges. Despite the comparable training loss observed between the two approaches, MIGA demonstrates a notably higher information correlation on the validation set. This superior performance on the validation set suggests that MIGA has a more robust capability to predict future profits, which is particularly significant in financial forecasting.

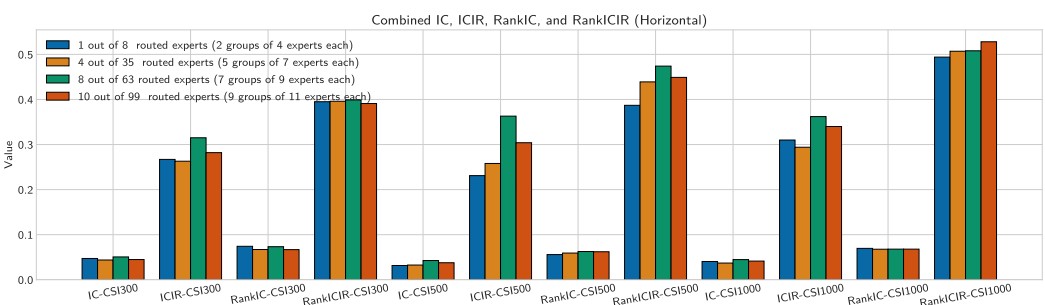

Figure 4: Comparison about different number of experts. The optimal setting of *8 out of 63 experts (7 groups of 9 experts each)* achieve the best result on 8/12 of the metrics.

Table 3: Impact of inner group attention. We compared the long-only portfolio results of 3 types of MIGA models with inner group attention or not. The results show that removing inner group attention led to a decline in model performance.

| Model | CSI300 | | CSI500 | | CSI1000 | |
|---|---|---|---|---|---|---|
| | AR | IR | AR | IR | AR | IR |
| MIGA-Conv | 0.21 | 1.56 | 0.16 | 0.94 | 0.14 | 0.64 |
| w/o inner group attention | 0.10 | 0.82 | 0.18 | 0.98 | 0.12 | 0.57 |
| MIGA-Rec | 0.06 | 0.46 | 0.10 | 0.58 | 0.17 | 0.85 |
| w/o inner group attention | 0.05 | 0.33 | 0.05 | 0.23 | 0.08 | 0.35 |
| MIGA-Attn | 0.19 | 1.44 | 0.08 | 0.46 | 0.12 | 0.53 |
| w/o inner group attention | 0.22 | 1.59 | 0.11 | 0.65 | 0.06 | 0.28 |

## 3.5 Ablation Study

**Scaling up the number of experts** To investigate the effect of expert aggregation size on model performance, we conducted an experiment using the MIGA architecture, with results illustrated in Figure 4. The findings indicate that increasing the number of experts leads to improved ICIR and RankICIR scores, suggesting that this approach can substantially enhance the stability of the model's prediction. To further explore the upper limit of the model's potential, we selected 8 out of 63 routed experts (organized into 7 groups of 9 experts each) for in-depth analysis in this paper.

**Impact of inner group attention** To evaluate the efficacy of inner group attention, we compare MIGA with a mixture of isolated experts that does not incorporate inner group attention. As shown in Table 3, MIGA outperforms the mixture of isolated experts and demonstrates a more balanced performance across the CSI300, CSI500, and CSI1000 benchmarks. This improvement can be attributed to the enhanced overall level of experts achieved through inner group attention, highlighting the effectiveness of inner group attention in facilitating more efficient and effective knowledge sharing among experts in Mixture of Experts (MoE) systems.

## 3.6 Analysis on Expert Specialization

To evaluate the expert specialization in MIGA, we conduct individual predictions on the test set using each expert in MIGA-Conv and compare their excess annual returns across various stock benchmarks and portfolios. As shown in Figure 5 and 6 (in Appendix A), the majority of experts demonstrate accurate predictions, with only 7 out of 63 experts failing to generate excess returns. This suggests that each expert has acquired sufficient predictive capabilities during MIGA's training. Furthermore, different experts exhibit varying levels of proficiency across different areas. For instance, in MIGA-Conv (Figure 5), Expert 3 achieves an impressive excess return of nearly 30% on CSI300 but fails to replicate this performance on CSI500 and CSI1000. In contrast, Expert 39 delivers a relatively high excess return on CSI500, but only mediocre results on CSI300 and CSI1000. This highlights the specialization of MIGA's experts for different types of stocks. Moreover, a similar specialization

was observed for the rise and fall of stocks. For example, also in MIGA-Conv (Figure 6), Expert 37 generates significantly higher excess returns for long positions compared to short positions, whereas Expert 4 excels in short positions, outperforming long positions.

## 4 RELATED WORK

**Stock market prediction** Recent advancements in stock market prediction have been propelled by end-to-end methods such as Temporal Convolutional Network (TCN) (Hewage et al., 2020), LSTM (Greff et al., 2016) and Transformer Vaswani et al. (2017). Although these end-to-end models were not originally designed for stock market forecasting, subsequent research (Ding et al., 2020; Nelson et al., 2017; Sun et al., 2023) has validated their effectiveness in this domain, which motivated us to propose MIGA and investigate its model structure.

**Mixture of Experts** The mixture-of-experts (MoE) approach has been widely adopted and extensively researched since its introduction over two decades ago (Jacobs et al., 1991). Many MoE models have achieved remarkable success in various challenging fields, including computer vision (Riquelme et al., 2021; Wang et al., 2021) and natural language processing (Shazeer et al., 2017b; Du et al., 2022). The effectiveness of this approach is further demonstrated by the success of MoE-based large language models, such as DeepSeekMoe (Dai et al., 2024) and Mixtral (Jiang et al., 2024). However, despite the broad applicability of MoE methods, a notable gap exists in the development of MoE frameworks specifically tailored to quantitative investment in stochastic financial markets. which inspire us to further explore it and propose MIGA.

## 5 CONCLUSION

This paper presents MIGA, a series of Mixture of Experts with Group Aggregation models that combines different expert for stock market prediction. Our approach demonstrates the potential of integrating Mixture of Experts framework in quantitative investment, enabling end-to-end models to tackle stochastic stock market. MIGA achieves state-of-the-art performance on 3 Chinese Stock Index benchmarks, substantially outperforming existing end-to-end forecasting approaches. Furthermore, we conduct systematic analysis of the benefits aspire for this work to provide valuable insights for future research, contributing to the development of more advanced real-world market solution.

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

# A EXPERT VISUALIZATION

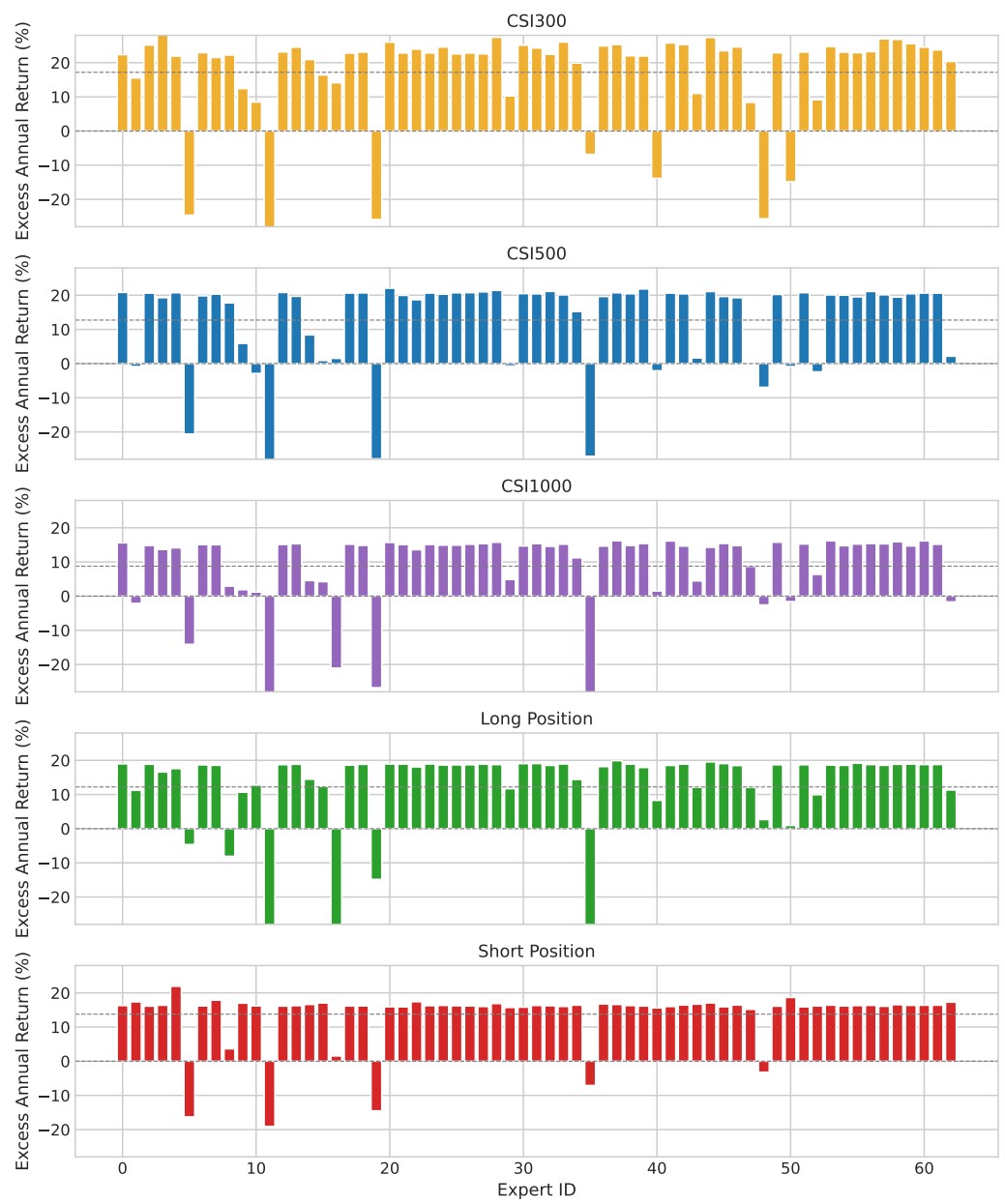

Figure 5: Stock specialization (top) and portfolio specialization (bottom) of MIGA-Conv.

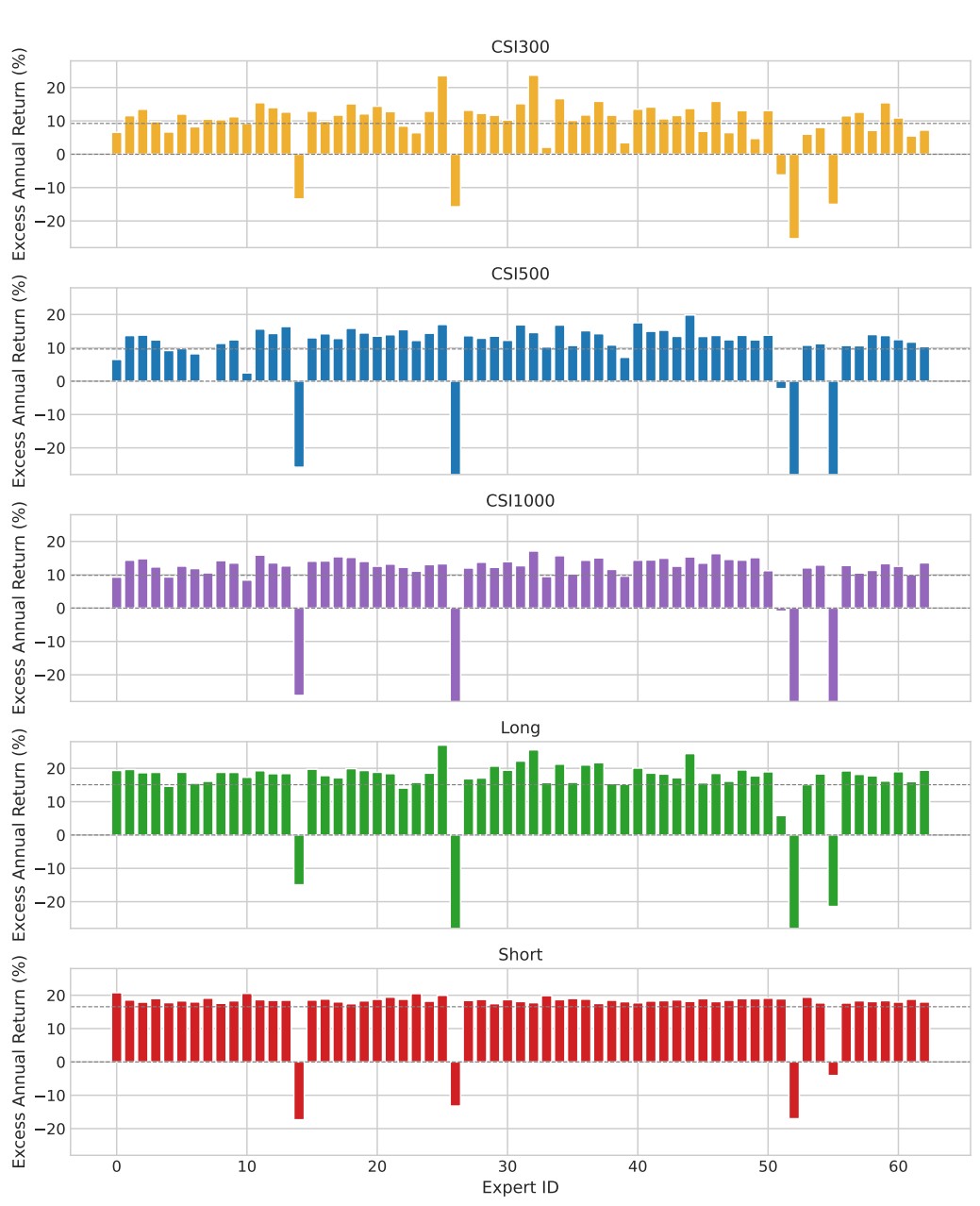

Figure 6: Stock specialization (top) and portfolio specialization (bottom) of MIGA-Rec.

