# OpenReview forum: "MIGA: Mixture-of-Experts with Group Aggregation for Stock Market Prediction"
_ICLR.cc/2025/Conference — ICLR 2025 Conference Withdrawn Submission_

### Official Review · Reviewer_K815 · 2024-10-30

**Soundness:** 3
**Presentation:** 3
**Contribution:** 1
**Rating:** 3
**Confidence:** 4

**Summary:**

This paper proposes MIGA for stock market prediction. To generate specialized predictions of various styles of stocks, the authors propose a Mixture of Expert framework which dynamically switching between specialized experts.

**Strengths:**

1. In the context of financial markets, stocks of different styles indeed exhibit significant variations in their features of stock factors. The problem to be solved is sound and makes sense to me.

2. The presentation is clear, especially Figure 2 clearly illustrates the framework of MIGA.

3. The choice of metrics is suitable and comprehensive. Annualized Return ratio presents the profit, Information ratio presents the risk, IR and RankIR presents the model power.

**Weaknesses:**

1. The motivation of the paper is quite similar to [1], which also leverages multiple experts to make an investment decision. Correct me if I'm wrong, the contributions of MIGA based on Alphamix are: (1) the router which filters out some of the experts. (2) the inner group aggregation module. I suppose the novelty of MIGA in the existence of Alphamix is limited. By the way, I recommend to include Alphamix as one of the baseline models.

2. It seems that statistics in Table 3 can't illustrate the effectiveness of the inner group attention, which is one of the new modules proposed. Some of the metrics are on par, some metrics even degrade. This ablation study doesn't provide a strong support to the main claim of the paper. The authors are suggested to report the statistical significance of Table 3.

3.  The related work section is short, and many works [2~5] in the field are missing.

[1] Sun et al. Mastering stock markets with efficient mixture of diversified trading experts.

[2] Feng et al. Enhancing stock movement prediction with adversarial training.

[3] Xu et al. HIST: A graph-based framework for stock trend forecasting via mining concept oriented shared information.

[4] Gao et al. DiffsFormer: A Diffusion Transformer on Stock Factor Augmentation.

[5] Huang et al. GENERATIVE LEARNING FOR FINANCIAL TIME SERIES WITH IRREGULAR AND SCALE-INVARIANT PATTERNS.

**Questions:**

What's the difference between AlphaMix and MIGA?

---

### Official Review · Reviewer_oY9q · 2024-11-04

**Soundness:** 1
**Presentation:** 2
**Contribution:** 2
**Rating:** 3
**Confidence:** 3

**Summary:**

The paper introduces a mixture of experts with a group aggregation framework, designed to improve stock market prediction accuracy. The work employs a dynamic switching mechanism between specialized style experts to capture the variations in stock styles and market trends. An innovation is introducing an inner group attention architecture, which promotes information sharing among experts within the same group.

**Strengths:**

1. The paper's combining multiple specialized experts with a group aggregation mechanism represents a novel approach, distinguishing this work from others in the field.
2. The paper conducts extensive experiments on three major Chinese Stock Index benchmarks (CSI300, CSI500, and CSI1000), providing a robust evaluation of the model's performance across different market conditions and stock types.
3. The model shows consistent improvements across various metrics, including ICIR and RankICIR, demonstrating its robustness and reliability.

**Weaknesses:**

1. The ablation study on scaling up the number of experts shows that increasing the number of experts has a minimal impact on the model's performance. This raises questions about the necessity and effectiveness of using a large number of experts in the framework.
2. The figure representations in the paper lack clarity. In particular, Figure 3 is only
briefly discussed in the text. There is no detailed explanation of why the performances of the models are different at the beginning of the training process while the training loss is similar.
3. The paper lacks a detailed discussion of the related works.

**Questions:**

1. Could the authors provide more insights into why increasing the number of experts does not significantly improve the model's performance?
2. Could the authors provide a more detailed explanation of the differences in model performance at the beginning of the training process, despite similar training losses? Additionally, how can the visualizations be improved to better communicate the training dynamics and performance metrics?
3. The paper mentions the use of inner group attention and the dynamic switching mechanism. Could the authors discuss the sensitivity of the model to hyperparameters related to these components? How do changes in these hyperparameters affect the model's performance?

---

### Official Review · Reviewer_ACY6 · 2024-11-04

**Soundness:** 3
**Presentation:** 3
**Contribution:** 2
**Rating:** 5
**Confidence:** 4

**Summary:**

This paper proposes a Mixture of Expert with Group Aggregation framework for stock market prediction. An inner group attention architecture is devised to enable experts of the same group to share information. The experiments on three benchmarks demonstrate the effectiveness of the proposal.

**Strengths:**

(1) The idea of the proposal seems sound.
(2) The paper is generally well-written and easy to follow.
(3) The performance surpasses the previous SOTA methods by a large margin.

**Weaknesses:**

(1) the major drawback is that no MOE baselines is compared in the experiments. And why the conventional MOE is not compared is not clear. If it can be directly used for stock market prediction, please make the comparison. Otherwise, please describe the challenges and how to overcome them.
(2) The novelty of the proposed inner group attention seems limited.
(3) The experiments are only performed on the Chinese stock datasets.

**Questions:**

What are the challenges when using the MoE approach in stock market prediction.
What is the advantage of the proposal when it is compared to other MoE methods.

---

### Note · Authors · 2024-11-24

I have read and agree with the venue's withdrawal policy on behalf of myself and my co-authors.